# Navigating Head and Neck Porocarcinoma: Systematic Review with Special Emphasis on Surgical Safety Margins

**DOI:** 10.3390/cancers16071264

**Published:** 2024-03-24

**Authors:** Alvija Kučinskaitė, Gintarė Ulianskaitė, Justinas Pamedys, Domantas Stundys

**Affiliations:** 1Faculty of Medicine, Vilnius University, 03101 Vilnius, Lithuania; alvija.kucinskaite@mf.stud.vu.lt; 2Institute of Clinical Medicine, Faculty of Medicine, Vilnius University, 03101 Vilnius, Lithuania; gintare.ulianskaite@santa.lt; 3National Centre of Pathology Affiliated to Vilnius University Hospital Santaros Klinikos, 08406 Vilnius, Lithuania; justinas.pamedys@vpc.lt

**Keywords:** eccrine porocarcinoma, malignant eccrine poroma, poroma, head and neck, wide local excision, surgical safety margins, minimal safety margins

## Abstract

**Simple Summary:**

Eccrine porocarcinoma poses challenges in diagnosis, as biopsy results can be misleading. Even if it is essential to surgically remove the suspected lesions, there is a lack of consensus regarding the appropriate surgical margins. In this study, we conducted a systematic review focusing on porocarcinoma cases in the head and neck region, encompassing a total of 20 cases which were documented with excision margins up to the present time. Our analysis revealed that surgical margins showed no significant variance based on age or specific anatomical regions, but rather correlated closely with tumour size. Despite thorough analysis, the rarity of the disease and the limited disclosure of safety margin details in case reports hindered our ability to define the minimum safety margins required for complete eccrine porocarcinoma surgical removal. Further research is warranted to address this gap in knowledge.

**Abstract:**

Eccrine porocarcinoma, sharing many features with other skin tumours, is diagnostically challenging. A conventional biopsy might be misleading and surgical excision becomes a primary diagnostic tool and a treatment method. However, the data on surgical safety margins are not consistent. We present a systematic review analysing the surgical margins of porocarcinoma in the head and neck area, which was conducted across the PubMed, Cochrane, and Web of Science databases including studies published from inception to November of 2023. In this systematic review, the PRISMA-ScR checklist was used, and a Cohen’s Kappa coefficient of 0.92 was applied, indicating very good agreement between reviewers. Out of 529 identified articles, 18 studies yielding 20 cases in total were selected for a thorough analysis. Nine (45%) cases were observed in the facial regions, eight (40%) on the scalp, and three (5%) on the neck. The primary treatment of choice was wide local excision with safety margins ranging from 3 to 22 mm (mean: 10.1). It demonstrated that surgical margins do not differ by age or anatomic regions, with the main point of reference being the tumour size. As observed, the bigger the tumour, the wider the safety margins were. However, the limited disclosure of surgical safety margins in analysed case reports impeded our ability to define the minimum safety margins. Further investigation and a consensus on recommended safety margins are required.

## 1. Introduction

Eccrine porocarcinoma (EPC) is a rare adnexal skin appendix tumour, arising from the intraepidermal ductal part of eccrine sweat glands which was first described in 1963 by Pinkus and Mehregan [1]. It accounts for about 0.005–0.01% of all skin tumours [2] and usually affects the elderly, with a mean age of 66 years and no gender predominance [3,4]. The most common locations are the lower extremities, head and neck (H&N), and trunk [5]. Due to the rarity of the tumour, the cause is unknown. It could arise de novo or develop from pre-existing lesions, such as poroma with a long latency period [5,6]. There is an increased risk of EPC for patients with a history of radiotherapy, chronic ultraviolet exposure, and immunosuppression [6,7].

The mean period between tumour development and diagnosis is 5 to 9 years [8]. Some articles report a prolonged period of inactivity of a prior lesion, followed by a sudden rapid growth. About 18 to 50 percent of EPC originates from benign eccrine poroma (EP) [2].

The tumour presents as an erythematous violaceous nodular lesion with or without ulceration, bleeding, or crusted scabs [8], which are nonspecific and commonly encountered features in malignant skin tumours. Furthermore, EPCs are usually well tolerated by patients until the bleeding or intolerable size causes difficulties. The most common dermoscopy features are atypical polymorphic vessels and milky-red globules [6,9]. EPC has also been reported to share dermoscopic characteristics with EP [10].

Diagnosing based on clinical features is inherently difficult due to their lack of specificity. The benign counterpart of EPC, eccrine poroma, is commonly called the “great imitator”. Therefore, relying solely on clinical examination and dermoscopy results in low diagnostic accuracy. Conventional biopsy is frequently performed to establish the correct diagnosis, although excisional biopsy should be prioritised to ensure an accurate diagnosis due to the risk of the tumour being a collision one [11,12,13].

The histological features of EPC encompass basaloid epithelial cells exhibiting ductal differentiation and cytologic pleomorphism. Additionally, ulceration and necrosis frequently manifest alongside heightened mitotic activity. Moreover, nuclear hyperchromia is a notable characteristic of EPC cells. Consequently, employing histological staining techniques such as EMA, PAS, CKs (AE1/AE3), p63, and Ki67, which have been reported as positive in 100% of cases, is imperative for ensuring accuracy and minimising the risk of misdiagnosis [2].

In select cases, sentinel lymph node biopsy should be performed, even in the absence of palpable lymph nodes. This procedure has an 81.3% success rate in identifying occult lymph node metastasis [2]. Further investigation with sentinel lymph node biopsy is highly indicated in specific scenarios, such as aggressive tumour types, lymphadenopathy, and histopathological criteria indicating a higher risk of metastasis (e.g., depth > 7 mm, lymph vascular invasion, and >14 mitoses) [14]. Standardised guidelines for managing EPC are lacking; thus, a multidisciplinary team approach is recommended.

While radiotherapy or chemotherapy may be used for disease control [6,15], wide local excision (WLE) is the primary treatment, with a debate of optimal surgical margins ranging from 2 mm to 3 cm [16,17]. In delicate localisations like the head and neck areas, >1.5 cm safety margins may be unfeasible. Precision in head and neck surgery is crucial in achieving optimal cosmetic outcomes by minimising scarring and disfigurement while preserving the function of critical structures. Mohs surgery (MMS) increasingly outperforms WLE in reducing the likelihood of tumour recurrence [18,19,20], though its availability may be limited to larger medical centres or regions with expertise. On the contrary, WLE can typically be performed in a broader range of healthcare facilities by various surgical specialists, including dermatologists, general surgeons, and plastic surgeons.

Despite tumour excision, it is known that about 20% of EPC recurs locally after surgery, with about 20% rate of regional metastasis and a mortality rate of 67% [2]. Approximately 22.3 percent of patients receive the diagnosis of metastatic disease, primarily involving regional lymph nodes (17%). The lesions on the lower extremities are considered to have the highest risk of metastasis; EPC is less likely to metastasize when located in the H&N areas. The most common areas of metastasis are the lymph nodes, lungs, liver, and brain [8].

Given the noted inconsistencies in data concerning adequate surgical safety margins, we present a systematic literature review focusing on previously described porocarcinoma cases in the head and neck region that have been documented with excision margins up to the present time.

## 2. Materials and Methods

The systematic review was conducted in adherence to the Preferred Reporting Items for Systematic Reviews and Meta-Analyses (PRISMA) guidelines [21]. The review protocol was registered at the Open Science Framework. The literature search was conducted across the PubMed, Cochrane, and Web of Science databases, including studies published from 1963 to November of 2023. The following search terms were used in adherence to Boolean logic: (eccrine porocarcinoma) OR (malignant eccrine poroma). The articles were evaluated by two reviewers (AK, DS); consensus regarding selection discrepancies was reached by consulting a third researcher (GU). The level of interrater agreement was determined by calculating the Kappa coefficient [22]. Values were interpreted as follows: ≤0.2 poor, 0.21–0.40 fair, 0.41–0.60 moderate, 0.61–0.80 good, and 0.81–1.00 as very good agreement.

### 2.1. Study Selection Criteria and Quality Assessment

The results were refined by following the established study inclusion criteria, established according to the PICOS principle:
P—Population: patients with porocarcinoma in H&N region.I—Intervention: surgical excision with specified safety margins.C—Comparison: pre- vs. post-surgical histopathological diagnosisO—Outcome: resection margin, recurrence, discrepancies in histopathological diagnosis.S—Study design: prospective/retrospective/case-control/cohort studies, randomised controlled trials, case reports, case series with sufficient single patient data.

The search was limited to articles in the English language with the free full-text article accessible through the extensive academic library network. The approach proposed by the JBI Critical Appraisal Checklist for case reports was employed to evaluate the quality of evidence and risk of bias [23]. The reports were rated on a 4-point scale concerning eight aspects. Each specific question was evaluated as follows: “yes”—1 point, “no/unclear”—0 points, “not applicable”—excluded from scoring. A score surpassing 5 was considered indicative of a valid case report.

### 2.2. Data Analysis

Pertinent individual patient information was extracted from the accepted articles. This information was then summarised and tabulated. According to their localisation, tumours were categorised into distinct regions following the Facial Aesthetic Unit Classification proposed by Fattahi [24]. Additionally, the tumours were classified into size groups: (1) <20 mm, (2) 20–40 mm, and (3) >40 mm. The groups for different tumour sizes were created by combining the information on basal and squamous cell carcinomas provided by the National Comprehensive Cancer Network^®^ [25].

The statistical analysis was performed using R Statistical Software (version 4.3.2; R Foundation for Statistical Computing, Vienna, Austria). Differences among groups in relation to other clinicopathologic parameters were analysed using the Kruskal–Wallis, Fisher’s exact, and Dunn’s tests.

## 3. Results

### 3.1. Literature Search

Initially, the search yielded 1366 articles. After removing duplicates, the results were reduced to 529, which were evaluated following the inclusion criteria. The Kappa coefficient value of 0.92 was applied, indicating very good agreement between raters. As many as 511 articles were excluded due to insufficient single patient data or undisclosed surgical safety margins. Out of the remaining 18, 20 cases were identified and subsequently included in this review (Figure 1). Patient data on age, sex, exact tumour location and characteristics, margins of surgical excision, follow-up period, and outcomes were recorded.

The surgical margins of H&N EPC were described in 18 studies, yielding 20 cases in total. The articles that were selected for a thorough analysis were published between 2004 and 2023. Among the cases, 14 were documented in individual case reports, while the remaining 6 were part of case series analyses. The selected articles were evaluated for methodological quality and risk of bias, following the guidelines of the JBI Critical Appraisal Checklist for case reports [23]. Since no patients encountered adverse events, the #7 criterion was changed to one focused on information regarding disease recurrence. The evaluation score for each article is presented in Table 1.

### 3.2. Patient Characteristics and Clinical Details

The mean patient age at the time of diagnosis was 66.85 years (range: 19–93). In total, there were 11 (55%) female and 9 (45%) male cases reported. The initial lesion biopsy was performed for nine cases with only four (44%) being consistent with the final pathological results. Five (56%) cases that appeared to have a mismatch between initial biopsy and postsurgical diagnosis were examined by employing either incisional or punch biopsy. The majority (85%) of tumours were singular EPCs, while three of them manifested as collision lesions: EPC + poroma [13], EPC + trichoblastoma [11], and EPC + basal cell carcinoma [12].

The primary treatment of choice was WLE with safety margins ranging from 3 to 22 mm (mean: 10.1 mm). Complete tumour resection was confirmed in 17 cases, with 2 that were not completely excised due to deep border involvement [12,32] and 1 that was not disclosed [33]. Among the patient cohort with R0, 11 (73%) displayed no signs of metastatic disease. All patients without any regional or distal lymph node involvement that underwent solely WLE with margins that fell within the previously mentioned range had the result of a notable 0% recurrence rate [7,11,13,26,29,31,34,35,37,39].

The follow-up period was disclosed in 16 (80%) of the cases, and information on recurrence was disclosed in 17 (85%) of the cases. Any absence of information was attributed to unintentional discrepancies in medical documentation. A detailed summary of the selected articles is presented in Table 1.

### 3.3. Descriptive Statistics by Anatomic Tumour Region and Size Group

The H&N tumours from the patient cohort were classified into six localisation groups: the scalp (8), forehead (1), upper and lower eyelid (1), cheek (3), nose (4) units, and neck region (3). The mean tumour size was 27.75 mm (range: 4–80 mm). The descriptive statistics by anatomic tumour region are presented in Table 2. The majority of our sample’s tumours were <20 mm (45%) or 20–40 mm (35%). The patients in the size group of >40 mm (20%) were the youngest (mean: 53 years). The biggest tumour size was observed in the neck region, with a mean value of 27 ± 29 mm. Our sample depicted that the scalp region tumours varied in size the most, ranging from 5 to 80 mm. Surgical safety margins were the smallest for tumours in the <20 mm group (mean: 8.6 mm ± 6.9 mm). The mean values of surgical safety margins tended to be almost twice as big for the cheek, nasal units, and scalp compared to other localisations. The descriptive statistics by tumour size group region are presented in Table 3.

Statistical hypothesis testing was performed in order to test whether tumour size (Table 2) and different anatomic regions (Table 3) could differ by age, gender, safety margins, and size. We used the Kruskal–Wallis rank sum test for continuous variables and Fisher’s exact test for categorical variables. We found that EPC localisation in our cohort in different H&N regions statistically significantly differed by age (*—*p*-value <0.05). The Dunn’s test for pairwise multiple comparisons did not find statistically significant pairs, but the *p*-value of age difference between the scalp region and cheek unit was 0.06. We consider this to be the case for explaining Kruskal–Wallis test results. As the hypothesis testing revealed that only age was statistically significantly different for H&N regions, the mean and standard deviation with minimum and maximum values were selected to present in the tables.

## 4. Discussion

Diagnosing EPC is currently challenging due to clinical, dermoscopic, and histologic features overlapping with other skin tumours. It is established that EPC can originate from pre-existing benign lesions like poromas [5,6,7,11]. Research by Robson et al. in a clinicopathologic study showed that about 18% of lesions displayed accompanying characteristics suggestive of a benign poroma [40]. Histopathological diagnosis of the suspected EPC lesion from a biopsy sample is further complicated due to a display of squamous differentiation, which could lead to misinterpretation, particularly in differentiating between EPC, basal, or squamous cell carcinomas. Due to this reason, it is believed that the prevalence of EPC may be underestimated [41].

Our analysis of the patient cohort revealed a 56% inconsistency when comparing the results of the initial biopsy with the subsequent pathological evaluation after complete surgical removal of the lesion. This cohort represents only a small fraction of all documented EPC cases in the literature, suggesting that the clinical disparity between the biopsy and post-surgical diagnoses may be even greater. Inadequate tissue sampling for histopathological examination and the absence of immunostaining could lead to different diagnoses. This carries the potential for an inaccurate initial diagnosis following a standard 3 mm punch biopsy, as only the benign part of the tumour could be provided for assessment. This becomes particularly problematic when the initial biopsy results indicate a benign lesion such as eccrine poroma, potentially resulting in the selection of less aggressive treatment options such as observation or electro destruction. To comprehensively evaluate a poroma-like lesion and ascertain whether it is benign or malignant, complete removal of the entire tumour for examination is necessary. The final diagnosis should not be confirmed only by the conventional punch or incisional biopsy. Consequently, we highly recommend that either shave biopsy or WLE be employed as the primary diagnostic and treatment method exclusively for poroma management.

Due to the rarity of EPC, there are no standardised treatment guidelines. A recent meta-analysis of 120 cases by Son Le et al. [20] revealed that in 92.5% of cases, surgical treatment is applied. In most cases (76.7%), WLE is performed, with the rest being MMS (15.8%) [20]. Belin et al. suggested that a choice of EPC treatment regimen could be made based on the growth pattern of the tumour. He proposed that the clinically suspected EPC should be initially excised with a 3 mm safety margin. After histopathological results confirm EPC, the subtype (pushing, infiltrative, pagetoid) should be clearly described. In the case of a pushing variant, no further surgery is needed; if the results verify the pagetoid or infiltrative subtype, additional surgery is required with a further 5 mm safety margin [16]. Even though MMS is a promising choice of treatment due to the accuracy of confirming clear surgical margins, it is expensive, time-consuming, and not as widely used as WLE.

While a conclusive diagnosis must primarily rely on histopathological findings, it is essential to be cautious. This is because EPC cells could also display squamous differentiation, a factor that may introduce further confusion and potentially lead to misinterpretation, particularly in differentiating between EPC, basal, or squamous cell carcinomas. Due to this reason, it is believed that the prevalence of EPC may be underestimated [41].

In the literature on EPC, most cases were documented to undergo “wide local excision” as the primary method of tumour treatment. Problematically, the definition of this term in the context of dermatooncosurgery varies among different authors. In general, WLE is understood to involve the margin ranging from 1 to 2 centimetres (10 to 20 mm) or more around the tumour. Considering that the average size of EPC lesions in the H&N region is 30.1 mm [41], there is a notable risk that surgical excision with margins of 10 mm and more may lead to severe deformities or necessitate extensive reconstructions. Therefore, it is of outmost importance to note the significance of minimal surgical margins for the tumours located in the H&N areas. The surgical margins for WLE in these regions should be enough for complete tumour resection but not more, which could cause severe unnecessary defects.

Our analysis of the literature reveals that surgical excision of H&N EPC typically involves margins ranging from 3 to 22 mm, with an average of 10.1 mm. This range may be safe for patients without any regional or distal lymph node involvement. The selection of safety margins remains consistent across different age groups and anatomical regions, primarily determined by the size of the tumour: the bigger the tumour, the wider the surgical margins.

While our studied patient cohort closely paralleled those previously documented in the literature in terms of patient demographics and clinical characteristics [2,3,4], it is crucial to emphasise the main limitations of the article: (a) the sample size was limited to only 20 patients, which underscores the representativeness of the cohort within the existing body of research, and the small sample size may not be sufficient to draw statistically significant conclusions regarding the surgical safety margins that would be just enough for complete resection (R0); (b) the systematic review relies on case report case series analysis due to the absence of randomised trials, which could explain the low prevalence of the tumour. This reason limited us in drawing a more confident conclusion; (c) we analysed only the tumours in the head and neck area, which makes the scope of the analysis narrower and more specific and does not provide information on surgical margins in the other body sites. Despite these limitations, our study still provides valuable insights and lays groundwork for further investigation into this critical aspect of surgical management. Complete resection of EPC would enable us to remove all visible traces of EPC-affected tissue, leaving no detectable remaining tumour cells behind, preventing further disease spread. This would result in improved outcomes and prognosis for the patient.

This systematic review examines both diagnostic and treatment aspects of EPC, highlighting the importance of examining the complete lesion and selecting precise surgical margins which would facilitate accurate diagnosis and enable the choice of more aggressive treatment approaches. Furthermore, the review provides insights into poroma diagnosis, proposing that WLE should serve not only as a treatment but also as a diagnostic tool. The strength of our analysis lies in its dedicated focus on surgical safety margins, particularly in the delicate areas of the face, scalp, and neck, where cosmetic outcomes greatly impact patients’ quality of life. Despite thorough analysis, the rarity of the disease and the limited disclosure of safety margin details in case reports hindered our ability to define the minimum safety margins required for complete surgical removal of EPC. Therefore, further research and consensus-building on precise safety margins for WLE are necessary to guide surgeons effectively.

## 5. Conclusions

The diagnosis of EPC might be challenging; therefore, we suggest a complete excision to establish the diagnosis to avoid inaccurate diagnosis. Furthermore, there are no consistent data on surgical margins of EPC in the face and neck area; therefore, further investigation is needed to establish the safety margins to increase the quality and safety of provided surgical treatment.

## Figures and Tables

**Figure 1 cancers-16-01264-f001:**
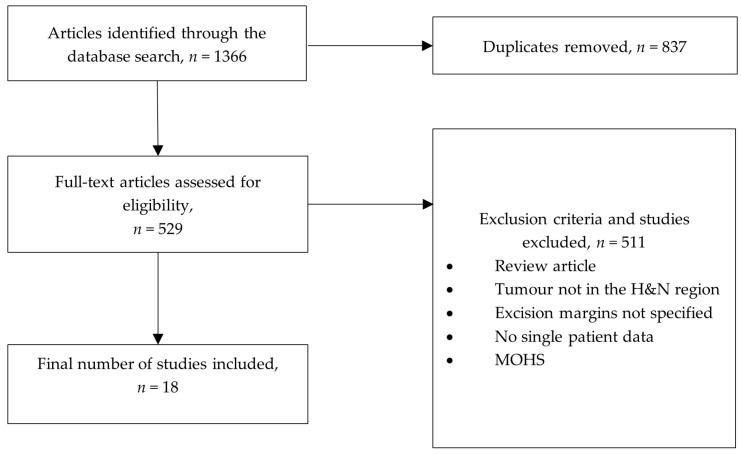
Article selection flow diagram.

**Table 1 cancers-16-01264-t001:** Detailed summary of the presented articles.

First Author	Study Type	JBI Score	Sex		Age	Time from Onset to Diagnosis	Localisation	Maximum Diameter of Tumour (mm)	Initial Biopsy (Type and Result)	Primary Treatment	Safety Margins, (mm)	Complete Resection	Histopathological Diagnosis Postsurgically	Lymph Node Involvement + Assessment Method	Additional Treatment	Follow-up (Months)	Recurrence
Arslan et al., 2004 [26]	Case report/letter to the editor	8	Male		75	3 years	Face (left ala nasi)	15	-	WLE	22	Yes	EPC	No, palpation	No	12	oNo
Kose et al., 2006 [7]	Case report	8	Female		52	20 years	Scalp (occipital region)	80	-	WLE	20	Yes	EPC	No, MRI	No	12	No
Luz et al., 2010 [27]	Case series	8	N1	Female	61	Mean 36 months [7–120 months]	Scalp	65	-	WLE	10	Yes	EPC	Yes, ND	Yes, radiation therapy	36	DOD
			N2	Male	58	Mean 36 months [7–120 months]	Neck	60	Squamous cell carcinoma—invasive clear-cell variant	WLE	10	Yes	EPC	Yes, ND	Yes, lymphadenectomy and radiation therapy	18	AWD
Kim et al., 2012 [28]	Case report	8	Male		52	15 years	Scalp (temporal area)	37	Eccrine poroma	WLE	20	Yes	EPC	No	Yes, radiation therapy	1	No
Nguyen et al., 2014 [29]	Case series	8	Female		54	ND	Neck (left superior region)	4	EPC	WLE	5	Yes	EPC	No, ND	No	44	No
Fujimura et al., 2014 [30]	Case report	8	Female		85	3 months	Right cheek	20	EPC	WLE	20	Yes	EPC	Yes, PET	CyberKnife for metastases	12	No
Mak et al., 2015 [31]	Case report	8	Female		60	ND	Face (right medial canthal region)	18	-	WLE	3	Yes	EPC	No, ND	No	12	No
Alcon et al., 2015 [13]	Case report/letter to the editor	8	Female		68	3 years	Scalp (occipital region)	20	EPC from preexisting EP	WLE	10	Yes	EPC from preexisting EP	No, palpation	No	12	No
Melgandi et al., 2016 [32]	Case report	8	Male		42	5 years	Scalp (right occipital area)	50	Eccrine poroma	WLE	10	No (Involved deep resection margin)	EPC	No, MRI	Yes, radiotherapy	12	No
Gomez-Zubiaur et al., 2017 [33]	Case report	7	Male		88	Mean 58, 43 months [1–480 months]	Neck	18	-	WLE	4 (mean)	ND	EPC	Yes, SLNB	No	12	NSPD
Ermertcan et al., 2018 [34]	Case report/letter to the editor	8	Female		87	6 months	Face (right cheek)	40	-	WLE	10	Yes	No	No, PET and US	No	6	No
Fukui et al., 2019 [35]	Case report	8	Female		66	3 months	Face (nose) (right lateral nose wall)	8	Squamous cell carcinoma	WLE	3	Yes	EPC	No, PET and palpation	No	9	No
Seo et al., 2019 [36]	Case report	8	Male		85	5 years	Cheek	10	-	WLE	5	Yes	EPC	Yes, palpation and US	Yes, radiotherapy	14	No
Mitchell et al., 2021 [11]	Case report	8	Male		58	ND	Scalp (left posterior parietal region)	30	Trichoblastoma and EPC	WLE	15	Yes	Trichoblastoma and EPC	No, SLNB	No	36	No
Seretis et al., 2022 [37]	Case series	7	Male		84	ND	Face (temple)	10	-	WLE	5	Yes	EPC	No, ND	No	ND (no data at follow up time)	No
Chouhan et al., 2023 [38]	Case report	7	Female		78	ND	Face (nose)	20	Squamous cell carcinoma	WLE	15	Yes	EPC	No, contrast CT	No	ND (no data on follow up)	ND (no data on recurrence)
Meriläinen et al., 2023 [39]	Case series	8	N1	Female	72	ND	Scalp	11	-	WLE	15 (mean)	Yes	EPC	No, US and SLNB	No	91	No
			N2	Male	19	ND	Scalp	5	-	WLE	15 (mean)	Yes	EPC	No, SLNB	No	130	No
Park et al., 2023 [12]	Case report	7	Female		93	15 years	Nose	34	-	WLE	5 (Involved deep resection margin)	No	EPC and basal cell carcinoma	No; MRI, CT, US, PET	No, observation due to age	ND (no data on follow up)	ND (no data on recurrence)

US—ultrasonography, MRI—magnetic resonance imaging, PET—positron emission tomography, CT—computed tomography, ND—not disclosed, NSPD—no single patient data, DOD—died of disease, AWD—alive with disease.

**Table 2 cancers-16-01264-t002:** Descriptive statistics by anatomic tumour region.

Anatomic Region (*n* = 20)						
Characteristic	Cheek Unit, *n* = 3	Forehead Unit, *n* = 1	Nasal Unit, *n* = 4	Neck Unit, *n* = 3	Scalp, *n* = 8	Upper and Lower Eyelid Unit, *n* = 1
Age, years (*)						
Mean ± (sd)	86 ± (1)	84	78 ± (11)	71 ± (15)	53 ± (17)	60
Min − Max	85–87	84	66–93	54–88	19–72	60
Gender, *n* (%)						
Female	2 (67%)	0 (0%)	3 (75%)	1 (33%)	4 (50%)	1 (100%)
Male	1 (33%)	1 (100%)	1 (25%)	2 (67%)	4 (50%)	0 (0%)
Safety margins, mm						
Mean ± (sd)	11.7 ± (7.6)	5.0	11.2 ± (8.9)	6.3 ± (3.2)	14.4 ± (4.2)	3.0
Min − Max	5.0–20.0	5.0	3.0–22.0	4.0–10.0	10.0–20.0	3.0
Size group, mm, *n* (%)						
<20	1 (33%)	1 (100%)	2 (50%)	2 (67%)	2 (25%)	1 (100%)
20–40	2 (67%)	0 (0%)	2 (50%)	0 (0%)	3 (38%)	0 (0%)
>40	0 (0%)	0 (0%)	0 (0%)	1 (33%)	3 (38%)	0 (0%)
Max tumour size, mm						
Mean ± (sd)	23 ± (15)	10	19 ± (11)	27 ± (29)	30 ± (24)	19
Min − Max	10–40	10	8–34	4–60	5–80	19

**Table 3 cancers-16-01264-t003:** Descriptive statistics by tumour size group.

Size Group (*n* = 20)			
Characteristic	<20 mm, *n* = 9	20–40 mm, *n* = 7	>40, *n* = 4
Anatomic region, *n* (%)			
Cheek unit	1 (11%)	2 (29%)	0 (0%)
Forehead unit	1 (11%)	0 (0%)	0 (0%)
Nasal unit	2 (22%)	2 (29%)	0 (0%)
Neck unit	2 (22%)	0 (0%)	1 (25%)
Scalp	2 (22%)	3 (43%)	3 (75%)
Upper and lower eyelid unit	1 (11%)	0 (0%)	0 (0%)
Age, years			
Mean ± (sd)	67 ± (17)	74 ± (16)	53 ± (8)
Min − Max	19–88	52–93	42–61
Gender, *n* (%)			
Female	4 (44%)	5 (71%)	2 (50%)
Male	5 (56%)	2 (29%)	2 (50%)
Safety margins, mm			
Mean ± (sd)	8.6 ± (6.9)	13.6 ± (5.6)	12.5 ± (5.0)
Min − Max	3.0–22.0	5.0–20.0	10.0–20.0
Follow-up period, months			
Mean ± (sd)	45 ± (48)	8 ± (5)	20 ± (11)
Min − Max	9–130	1–12	12–36

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
