# Peer review of "Navigating Head and Neck Porocarcinoma: Systematic Review with Special Emphasis on Surgical Safety Margins"

_cancers, 2024, doi:10.3390/cancers16071264_

Round 1

Reviewer 1 Report

Comments and Suggestions for Authors

This is an interesting article on navigating head and neck porocarcinoma.

I have certain suggestions to enhance the manuscript:

1) The introduction should be shortened.

2) Literature search should be performed to more databases, preferably three

3) The kappa-coefficient should be estimated to assess the level of agreement between the reviewers

4) In Figure 1 the number of included studies is not clear

5) There is no assessment of quality of the included studies along with assessment of the publication bias

6) In table the drop-out rate of patients during long-term follow-up should be demonstrated regarding each study 

7) A clear paragraph describing all the potential limitations should be added.

Comments on the Quality of English Language

The quality of English language is adequate.

Author Response

Dear Reviewer,

Thank you for your time while reviewing our manuscript. We very much appreciate your insight. Taking into account your comments we have edited the manuscript and reflected your comments.

We have shortened the introduction, omitting excessive clinical description of this rare disease, its dermoscopical and histological evaluation as well as some thoughts on treatment methods.

In addition to PubMed, we have included Cochrane and Web of Science databases to our literature analysis, yielding 1366 articles in our initial pool. The newly included articles were reanalysed in the same approach by two reviewers as described in our methods. No new reports were found which could be included into statistical analysis.

We appreciate your sound insight about the Kappa Coefficient. We have done that in order to assess the agreement between the reviewers and calculated its value to be 0.92, which indicates very good agreement between the raters.

We have increased the font size and adjusted the visibility of Figure 1 making sure all the numbers are well readable.

To assess the methodological quality and publication bias we have chosen the guidelines of JBI Critical Appraisal Checklist for case reports. As all of our included articles have surpassed the score of 5, they were considered to be valid for our systematic review. We have included the JBI score of each study in Table 1.

As the follow-up period was clearly disclosed in 16 (80%) and information on recurrence in 17 (85%) articles of the cases we have appended this information to Table 1.

We have underlined the potential limitations in greater details in the separate paragraph of the discussion.

Once again thank you for your review!

Sincerely,

Domantas Stundys - on behalf of all the authors

Reviewer 2 Report

Comments and Suggestions for Authors

This is a systematic review dealing with a rare and intriguing cancer (Porocarcinoma); Authors focused their research on the adequate surgical excision modality needed to achieve safety free margins. 

As stated in the discussion, the main limitation of this study is the small sample size (only 20 cases reported) that makes difficult to draw conclusions about the validity of the oncological outcomes.  Moreover, the follow-up period reported from the presented articles is too short (= or < 12 months in many cases or not disclosed in three cases) to have a clear evidence of the recurrence rate and prognosis.    

-Authors wrote that "it is of outmost importance to note the significance of minimal surgical margins for the tumors located in H-N areas":  this is obviously a correct observation in general terms in order to limit severe cosmetic  deformities for patients. However, we must consider the severe prognosis of this rare cancer characterized by a insidious clinical behavior and late diagnoses in most caases.    Porocarcinomas should be appropriately treated in large medical centers with expertise in MOHS surgery that is the modality of choice to eradicate the tumour in  adequate safe free margins.   

-Because of the limited sample size of the cases analysis no definitive conclusions can be drawn about the optimal surgical margins to be obtained in the treatment of porocarcinoma. However, this review highlights an important clinical and diagnostic topic regarding a rare and insidious  skin cancer.

The references are appropriate and the tables are exhaustive. 

-Question to the Authors: how long was the period between suspected tumoral development and confirmed pathological diagnosis in the cases presented (if data available)? 

Author Response

Dear Reviewer,

Thank you for taking your time to review our manuscript. We appreciate your comments as an encouragement to further continue our research in this field. As our systematic review encompasses only 20 cases it is obviously complicated to draw sound clinical conclusions. Moreover, we have additionally underlined the potential limitations in the separate paragraph of the discussion.

We have also included the separate column in Table 1 answering your question about the time interval from the onset to diagnosis of the disease – it was not clearly reported in 7 cases and greatly varied in the other ones (from 3 months to 20 years)

Once again thank you for your review.

Sincerely,

Domantas Stundys - on behalf of all the authors

Round 2

Reviewer 1 Report

Comments and Suggestions for Authors

I believe that the manuscript has been adequately revised.